# Improving outcomes for donation after circulatory death kidney transplantation: Science of the times

Michèle J. C. de Kok[1], Alexander F. M. Schaapherder[1], Ian P. J. Alwayn [1], Frederike J. Bemelman[2], Jacqueline van de Wetering[3], Arjan D. van Zuilen[4], Maarten H. L. Christiaans[5], Marije C. Baas[6], Azam S. Nurmohamed [7], Stefan P. Berger[8], Esther Bastiaannet[9], Rutger J. Ploeg[1,10], Aiko P. J. de Vries[11], Jan H. N. Lindeman [1]*

1 Department of Surgery and Leiden Transplant Center, Leiden University Medical Center, Leiden, The Netherlands, 2 Department of Internal Medicine (Nephrology), Amsterdam UMC, Academic Medical Center, Amsterdam, The Netherlands, 3 Department of Internal Medicine (Nephrology), Erasmus University Medical Center, Rotterdam, The Netherlands, 4 Department of Internal Medicine (Nephrology), University Medical Center Utrecht, Utrecht, The Netherlands, 5 Department of Internal Medicine (Nephrology), Maastricht University Medical Center, Maastricht, The Netherlands, 6 Department of Internal Medicine (Nephrology), Radboud University Medical Center, Nijmegen, The Netherlands, 7 Department of Internal Medicine (Nephrology), Amsterdam UMC, VU Medical Center, Amsterdam, The Netherlands, 8 Department of Internal Medicine (Nephrology), University Medical Center Groningen, Groningen, The Netherlands, 9 Department of Surgery, Leiden University Medical Center, Leiden, The Netherlands, 10 Nuffield Department of Surgical Sciences, University of Oxford, Oxford, United Kingdom, 11 Division of Nephrology, Department of Internal Medicine and Leiden Transplant Center, Leiden University Medical Center, Leiden, The Netherlands

* lindeman@lumc.nl

**Data Availability Statement:** Because the data contain potentially identifying and sensitive patient information, they cannot be made publicly available. Access to the data can be requested

## Abstract

The use of kidneys donated after circulatory death (DCD) remains controversial due to concerns with regard to high incidences of early graft loss, delayed graft function (DGF), and impaired graft survival. As these concerns are mainly based on data from historical cohorts, they are prone to time-related effects and may therefore not apply to the current timeframe. To assess the impact of time on outcomes, we performed a time-dependent comparative analysis of outcomes of DCD and donation after brain death (DBD) kidney transplantations. Data of all 11,415 deceased-donor kidney transplantations performed in The Netherlands between 1990–2018 were collected. Based on the incidences of early graft loss, two eras were defined (1998–2008 [n = 3,499] and 2008–2018 [n = 3,781]), and potential time-related effects on outcomes evaluated. Multivariate analyses were applied to examine associations between donor type and outcomes. Interaction tests were used to explore presence of effect modification. Results show clear time-related effects on posttransplant outcomes. The 1998–2008 interval showed compromised outcomes for DCD procedures (higher incidences of DGF and early graft loss, impaired 1-year renal function, and inferior graft survival), whereas DBD and DCD outcome equivalence was observed for the 2008–2018 interval. This occurred despite persistently high incidences of DGF in DCD grafts, and more adverse recipient and donor risk profiles (recipients were 6 years older and the KDRI increased from 1.23 to 1.39 and from 1.35 to 1.49 for DBD and DCD donors). In contrast, the median cold ischaemic period decreased from 20 to 15 hours. This national study shows major improvements in outcomes of transplanted DCD kidneys over time. The time-

through the Dutch Transplant Foundation (Nederlandse Transplantatie Stichting) (info@transplantatiestichting.nl).

**Funding:** The authors received no specific funding for this work.

**Competing interests:** The authors have declared that no competing interests exist.

dependent shift underpins that kidney transplantation has come of age and DCD results are nowadays comparable to DBD transplants. It also calls for careful interpretation of conclusions based on historical cohorts, and emphasises that retrospective studies should correct for time-related effects.

## Introduction

In the past decades, organs retrieved from donation after brain death (DBD) donors have provided the majority of solid organ transplants globally. Due to the medical success of transplantation as an effective therapy for patients with end stage organ failure, the increased need of donor organs created a persistent shortage which has resulted in the death of many patients while waiting for a transplant.

For many years now, kidneys donated after circulatory death (DCD) have been proposed as an effective means of addressing this severe organ shortage [1, 2]. Despite emerging reports indicating that mid-term and long-term outcomes of DCD procedures are better than commonly thought [3–5], only some countries have fully embraced this opportunity [6]. For various reasons, others have been reluctant or even outspoken adverse towards the introduction of a controlled DCD programme that could alleviate the shortage and save many lives within a healthcare system [7, 8]. While for some countries reasons to not or only slowly allow DCD programmes relate to ethical issues, legal restrictions or logistical concerns [9], for the majority of countries the reticent attitude generally reflects medical concerns that are based on reported high incidences of early graft loss, delayed graft function (DGF), and an assumed impaired graft survival for DCD kidneys.

Since the concerns regarding the inferior DCD outcomes are mainly based on historical analyses, they are prone to time-related effects as time-varying confounding and effect modification by time [10, 11]. Time-varying confounding is the phenomenon that the values of confounding variables, such as donor and recipient age, change over time [10]. Effect modification by time occurs when *the effect* of donor type on *outcomes* is modified by time (e.g. due to changes in procedural characteristics and/or medical decision-making over time) [11]. Therefore, assumptions as regards inferior outcomes of DCD procedures may not apply anymore to our current timeframe.

To test whether conclusions with regard to the outcomes of DCD kidney transplant procedures are influenced by time, and to objectify the current results achieved when utilising DCD donor kidneys, we performed a longitudinal time-dependent comparative analysis of the outcomes of DBD and DCD kidney transplant procedures performed in The Netherlands, as country with a longstanding tradition of the use of DCD donor kidneys.

## Materials and methods

### Patient population

This national outcome evaluation was approved by the Ethics Committee of the Leiden University Medical Center, and the clinical and research activities being reported are consistent with the Principles of the Declaration of Istanbul as outlined in the 'Declaration of Istanbul on Organ Trafficking and Transplant Tourism'. Data was fully anonymized prior to access and analysis.

In this study, we collected data of all 11,415 deceased-donor kidney transplant procedures performed in The Netherlands between 1990 and 2018. Combined organ procedures (n = 635), procedures with grafts donated after uncontrolled circulatory death (i.e. Maastricht Category I: dead on arrival and II: unsuccessful resuscitation) (n = 212), and procedures in recipients younger than 12 years old (n = 261) were excluded.

To explore a possible time-related effect, the incidence of early graft loss was mapped for the years 1990 to 2018. In this analysis, only primary kidney transplant procedures (n = 8,511) were included since early graft loss after re-transplantation is potentially interfered with accumulation of recipient-related risk factors [12]. Based on the early graft loss incidences, two timeframes were defined for the time-dependent comparative analysis. This analysis included all (primary transplantations and re-transplantations) transplantations performed between 1998–2008 (n = 3,499) and 2008–2018 (n = 3,781).

Data was retrieved from the Dutch National Organ Transplant Registry, which is a mandatory registry that contains granular data of all eight Dutch kidney transplant centres.

## Definitions

Early graft loss was defined as graft loss within 90 days after transplantation. Patients who died within 90 days after transplantation with a functioning graft were not considered as early graft loss recipients. DGF was defined as the need for dialysis in the first postoperative week(s). The Modification of Diet in Renal Disease (MDRD) equation was used to estimate the glomerular filtration rate (eGFR) in the recipient. The non-scaled, donor-only version of the Kidney Donor Risk Index (KDRI) was calculated as described by Rao et al. [13]. The following definitions were used for ischaemic periods of the donor kidneys. The first warm ischaemic period is the time following the no touch period after circulatory arrest and asystole in the DCD donor, until cold flush-out in the donor is commenced. The cold ischaemic period is the time from start of cold flush-out until the start of the vascular anastomosis in the recipient. The graft anastomosis time is defined as the time from kidney removal from static cold storage or hypothermic machine perfusion until reperfusion in the recipient.

## Data analysis

IBM SPSS Statistics 23.0 (IBM Corp., Armonk, NY, USA) was used for statistical analysis. Comparisons between DBD and DCD procedures were performed using the independent t-test for normal-distributed data, the Mann-Whitney rank test for non-parametric data, and the Chi-Square test for categorical data.

The KDRI was reported to facilitate comparison of the Dutch donor cohort with that of other countries. However, in the Dutch National Organ Transplant Registry donor hypertension and diabetes—which are included in the KDRI—are only registered from 2000 respectively 2002 onwards. As such, there was a high proportion of missing data for the 1998–2018 interval (26.6% for diabetes and 10.0% for hypertension) and multiple imputation of missing data of variables included in the KDRI was applied.

Logistic and linear regression analyses were used to examine the association between outcomes (DGF, early graft loss and 1-year eGFR) and donor type. Cox proportional hazards analyses were performed to evaluate differences in patient survival and death censored graft survival. All the multivariate models were adjusted for variables statistically relevant (p-value <0.10) in the univariate analysis (S1 Table). To avoid overcorrection the KDRI was not included in the multivariate models (as the KDRI also comprises donor age and donor type which are already included separately in the univariate and multivariate analyses). Also the type of preservation solution was not included as the inter-relationship between variables (the

selection of preservation solution depends on donor type) would substantially impact the validity of the model. Results are represented as beta coefficient (β), Odds Ratio (OR) or Hazard Ratio (HR) with the corresponding 95% Confidence Interval (CI).

An interaction (Wald) test was used to explore the presence of effect modification by time. In other words, to test whether the effect of donor type on outcomes is modified by time. To specifically determine the association ($R^2$) between KDRI and 5-year graft survival, logistic regression analysis was performed using the data from 2002 to 2018 (non-imputed). P-values <0.05 were considered statistically significant.

## Results

To explore a possible time-dependent effect on transplant outcomes, we mapped the incidence of early graft loss, as an unambiguous outcome parameter, for the years 1990 to 2018 in The Netherlands (Fig 1). Analyses of 5,895 DBD and 2,616 DCD primary kidney transplants performed in this period indicated clear time-related effects with 1998 (DBD) and 2008 (DCD) as clear transition years, after which the incidence of early graft loss dropped and stabilized at an incidence of approximately 6% (Fig 1).

Based on the transition years for early graft loss, two timeframes (1998–2008 and 2008–2018) were defined, and the outcomes of all 7,280 DBD and DCD procedures performed in these periods were compared accordingly (Tables 1 and 2). This comparison showed a marked increase in donor and recipient age, and in KDRI over time: donors and recipients were respectively 6.5 and 6 years older in the recent (2008–2018) timeframe and the KDRI increased from 1.23 to 1.39 for DBD donors and from 1.35 to 1.49 for DCD donors (Table 1). Although the KDRI for this Dutch cohort was significantly associated with 5-year graft survival (p<0.001), the Nagelkerke $R^2$ for the KDRI and 5-year graft loss was only 1.7% suggesting a limited impact of donor characteristics on graft survival.

In contrast to the increase in donor and recipient age over time, there was a clear decrease in the cold ischaemic period from a mean value of 20 hours in the 1998–2008 era to approximately 15 hours in the 2008–2018 era (Table 1). Also the proportion of procedures with excessive cold ischaemic periods (>24 hours) substantially decreased over time (from 25.7% to 7.8% and from 20.1% to 3.2% for DBD respectively DCD donors).

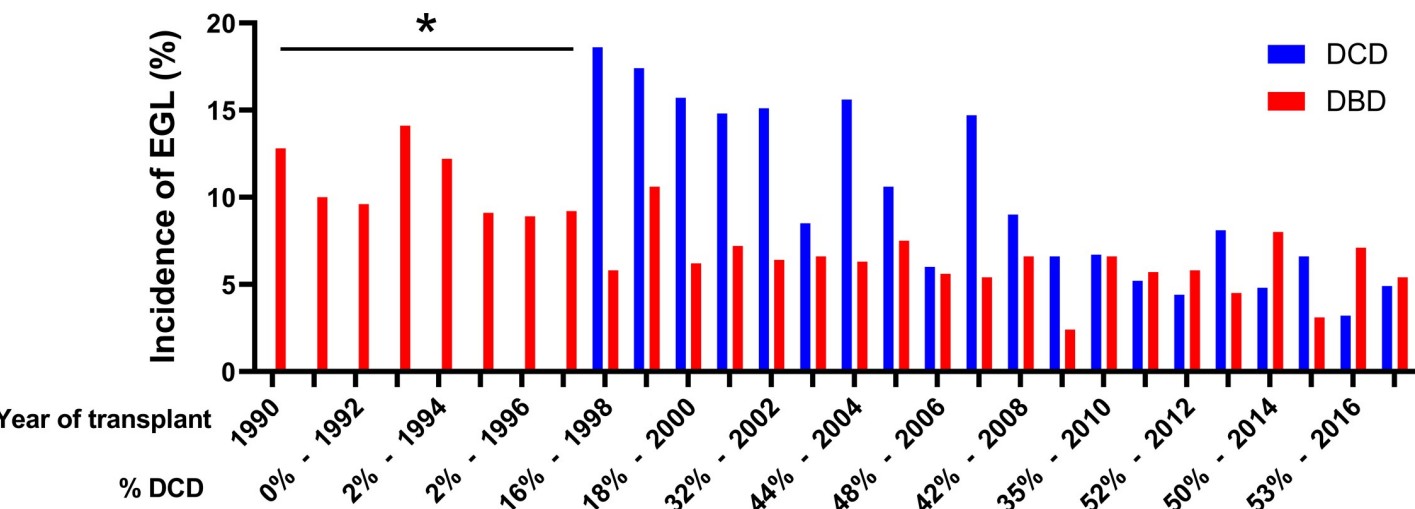

**Fig 1. Time-related incidences of early graft loss in 8,511 primary kidney transplant recipients according to deceased donor type in The Netherlands.** * The small number of DCD kidney transplant procedures performed in these years (n < 15), does not justify adequate power for analysis. DBD, donation after brain death; DCD, donation after circulatory death; EGL, early graft loss.

**Table 1. Baseline characteristics of deceased-donor kidney transplants in The Netherlands for the 1998–2008 and the 2008–2018 interval.**

| | | 01/1998–12/2007 | | | 01/2008–12/2017 | | | |
| | | DBD | DCD | p-value | DBD | DCD | p-value | p for interaction |
| | | n = 2361 (67.5%) | n = 1138 (32.5%) | | n = 1929 (51.0%) | n = 1852 (49.0%) | | |
| BASELNIE | Donor age | 46.2 ± 15.3 | 45.4 ± 15.4 | 0.16 | 52.7 ± 14.6 | 51.6 ± 14.5 | 0.03 | <0.001 |
| | Recipient age | 48.6 ± 14.3 | 50.7 ± 13.3 | <0.001 | 55.0 ± 14.2 | 55.9 ± 12.9 | 0.03 | <0.001 |
| | Time on dialysis (years) | 4.1 ± 2.5 | 4.2 ± 2.0 | 0.24 | 3.8 ± 2.5 | 3.6 ± 2.1 | 0.001 | <0.001 |
| | First warm ischaemic period (min.) | NA | 19.0 [15.0–25.0] | - | NA | 16.0 [13.0–19.0] | - | - |
| | Cold ischaemic period (hours) | 20.0 [15.6–24.6] | 20.0 [16.0–23.8] | 0.32 | 15.4 [11.8–19.8] | 14.0 [11.5–17.6] | <0.001 | <0.001 |
| | ≤24 hours | 1705 (74.3%) | 889 (79.9%) | | 1589 (92.2%) | 1575 (96.8%) | | |
| | >24 hours | 589 (25.7%) | 224 (20.1%) | | 135 (7.8%) | 52 (3.2%) | | |
| | Graft anastomosis time (min.) | 34.0 [27.0–41.0] | 33.0 [27.0–40.0] | 0.32 | 33.0 [26.0–41.0] | 32.0 [25.0–40.0] | 0.01 | 0.01 |
| | KDRI [1] | 1.23 [1.0–1.5] | 1.35 [1.1–1.6] | <0.001 | 1.39 [1.1–1.7] | 1.49 [1.2–1.8] | <0.001 | <0.001 |
| | Preservation solution | | | <0.001 | | | <0.001 | <0.001 |
| | - HTK | 325 (13.9%) | 967 (86.1%) | | 463 (24.4%) | 719 (39.3%) | | |
| | - University of Wisconsin | 2001 (85.4%) | 152 (13.5%) | | 1383 (72.8%) | 1058 (57.8%) | | |
| | - Other | 17 (0.7%) | 4 (0.4%) | | 55 (2.9%) | 53 (2.9%) | | |

Data are respectively presented as mean ± standard deviation, as number (%), as median [interquartile range] or as beta coefficient (β), odds ratio or hazard ratio with the corresponding 95% confidence interval.

[1] Multiple imputation was applied for missing data of variables included in the KDRI.

DBD, donation after brain death; DCD, donation after circulatory death; HTK, histidine-tryptophan-ketoglutarate; KDRI, kidney donor risk index; min., minutes; NA, not applicable.

Furthermore, the type of preservation solution changed over time. Whereas histidine-tryptophan-ketoglutarate (HTK) solution was most common for DCD donor kidneys between 1998 and 2008 (86.1%), University of Wisconsin solution was most commonly used between 2008 and 2018 (57.8%) (Table 1).

The comparative outcome analysis showed clear time-related effects (Table 2). Whereas the 1998–2008 era associates with compromised outcomes for DCD procedures in comparison to DBD procedures (i.e. higher incidences of DGF and early graft loss, impaired 1-year renal function in patients without DGF, and inferior 1- and 5-year graft survival rates), the 2008–2018 era shows outcome equivalence for DBD and DCD procedures. Only the incidence of DGF for DCD grafts remained high in the current era (Table 2), but this did not impact graft and patient survival. Yet, for both timeframes and for both donor types, DGF resulted in a 15% reduced 1-year renal function (Table 2).

To explore whether the shift towards outcome equivalence reflects effect modification by time, an interaction (Wald) test was performed. As expected, the interaction test confirmed a significant difference in the effect of donor type on outcomes (DGF, early graft loss, 1- and 5-year graft survival) between the two time eras (p for interaction: <0.001, 0.002, 0.001 and <0.001, respectively) (Table 2).

## Discussion

This national study demonstrates a major improvement in outcomes of transplanted DCD kidneys over time. Whereas the 1998–2008 era associates with inferior outcomes for DCD kidney transplant procedures, the 2008–2018 era shows outcome equivalence between DBD and DCD kidney transplants. This shift underpins that DCD kidney transplantation has come of

**Table 2. Deceased-donor kidney transplant outcomes in The Netherlands for the 1998–2008 and the 2008–2018 interval.**

| | | 01/1998–12/2007 | | | 01/2008–12/2017 | | | |
|---|---|---|---|---|---|---|---|---|
| | | DBD | DCD | p-value | DBD | DCD | p-value | p for interaction |
| | | n = 2361 (67.5%) | n = 1138 (32.5%) | | n = 1929 (51.0%) | n = 1852 (49.0%) | | |
| OUTCOME | Delayed graft function | 420 (17.8%) | 511 (44.9%) | <0.001 | 321 (16.6%) | 736 (39.7%) | <0.001 | |
| | Crude OR (95% CI) | Ref. | 4.04 (3.42–4.78) | <0.001 | Ref. | 3.85 (3.28–4.53) | <0.001 | <0.001 |
| | Adjusted OR (95% CI) | Ref. | 4.17 (3.45–5.04) | <0.001 | Ref. | 4.78 (3.99–5.70) | <0.001 | |
| | Early graft loss (<day 90) | 194 (8.2%) | 151 (13.3%) | <0.001 | 110 (5.7%) | 114 (6.2%) | 0.56 | |
| | Crude OR (95% CI) | Ref. | 1.71 (1.36–2.14) | <0.001 | Ref. | 1.09 (0.83–1.42) | 0.56 | 0.002 |
| | Adjusted OR (95% CI) | Ref. | 1.77 (1.40–2.23) | <0.001 | Ref. | 1.24 (0.92–1.68) | 0.16 | |
| | - Primary non-function | 44 (22.7%) | 58 (38.4%) | | 42 (38.2%) | 43 (37.7%) | | |
| | - Rejection | 58 (29.9%) | 25 (16.6%) | | 23 (20.9%) | 20 (17.5%) | | |
| | - Thrombosis or infarction | 38 (19.6%) | 36 (23.8%) | | 14 (12.7) | 24 (21.1%) | | |
| | - Other | 54 (27.8%) | 32 (21.2%) | | 31 (28.2%) | 27 (23.7%) | | |
| | 1-year eGFR DGF - | 52.6 ± 19.7 | 49.5 ± 18.1 | 0.02 | 51.5 ± 19.7 | 52.3 ± 20.0 | 0.44 | |
| | Crude β (95% CI) | Ref. | -3.07 (-5.67 - -0.47) | 0.02 | Ref. | 0.84 (-1.29–2.96) | 0.44 | 0.93 |
| | Adjusted β (95% CI) | Ref. | -4.21 (-6.57 - -1.86) | <0.001 | Ref. | -0.11 (-2.04–1.82) | 0.91 | |
| | 1-year eGFR DGF + | 44.1 ± 19.0 | 44.4 ± 18.4 | 0.81 | 44.8 ± 18.9 | 44.6 ± 17.5 | 0.89 | |
| | Crude β (95% CI) | Ref. | 0.32 (-2.19–2.82) | 0.81 | Ref. | -0.18 (-2.81–2.45) | 0.89 | 0.70 |
| | Adjusted β (95% CI) | Ref. | -0.69 (-3.02–1.63) | 0.56 | Ref. | -1.63 (-4.05–0.79) | 0.19 | |
| | 1-year graft loss | (10.9%) | (15.0%) | | (8.0%) | (7.9%) | | |
| | Crude HR (95% CI) | 1.0 | 1.41 (1.16–1.71) | 0.001 | 1.0 | 0.98 (0.78–1.23) | 0.84 | 0.001 |
| | Adjusted H R (95% CI) | 1.0 | 1.45 (1.19–1.77) | <0.001 | 1.0 | 1.09 (0.85–1.40) | 0.49 | |
| | 5-year graft loss | (20.0%) | (22.8%) | | (14.0%) | (13.6%) | | |
| | Crude HR (95% CI) | 1.0 | 1.18 (1.01–1.37) | 0.04 | 1.0 | 0.96 (0.80–1.15) | 0.66 | <0.001 |
| | Adjusted HR (95% CI) | 1.0 | 1.25 (1.07–1.46) | 0.01 | 1.0 | 1.04 (0.85–1.27) | 0.69 | |
| | 1-year patient survival | (95.0%) | (94.4%) | | (95.2%) | (94.8%) | | |
| | Crude HR (95% CI) | 1.0 | 1.14 (0.84–1.54) | 0.41 | 1.0 | 1.08 (0.82–1.43) | 0.58 | 0.60 |
| | Adjusted HR (95% CI) | 1.0 | 1.08 (0.80–1.46) | 0.62 | 1.0 | 1.08 (0.81–1.42) | 0.61 | |
| | 5-year patient survival | (82.8%) | (82.4%) | | (82.3%) | (82.5%) | | |
| | Crude HR (95% CI) | 1.0 | 1.04 (0.88–1.23) | 0.66 | 1.0 | 1.00 (0.85–1.17) | 0.96 | 0.81 |
| | Adjusted HR (95% CI) | 1.0 | 1.08 (0.89–1.30) | 0.44 | 1.0 | 1.02 (0.85–1.22) | 0.83 | |

Data are respectively presented as mean ± standard deviation, or as number (%), and beta coefficient (β), odds ratio or hazard ratio with the corresponding 95% confidence interval.

95% CI, 95% confidence interval; DBD, donation after brain death; DCD, donation after circulatory death; DGF, delayed graft function; eGFR, estimated glomerular filtration rate; HR, hazard ratio; min., minutes; OR, odds ratio.

age and results are nowadays comparable to DBD kidney transplants. It also emphasises that conclusions based on retrospective data (i.e. based on timeframes in which outcomes of DCD procedures were inferior to DBD procedures) are interfered by time-varying confounding and effect modification, and are therefore no longer justified.

Patient- and graft survival equivalence for DBD and DCD procedures occurred despite a persistent high incidence of DGF in DCD grafts. This apparent paradox can be explained by differential impacts of DGF on DBD and DCD outcomes, with a negligible impact of DGF on patient- and graft survival in recipients with DCD grafts [4, 14]. A phenomenon that presumably relates to donor type-specific molecular differences in organ resilience [14].

A clear, univocal explanation for the improved outcomes is missing. In the context of more adverse donor and recipient risk profiles in the more recent timeframe, the improvement in

DCD outcomes over time presumably involves a complex interplay of factors that includes optimized surgical procedures and immunosuppressive regimens, altered organ preservation techniques, and enhanced transport logistics [15–19]. Certainly, a significant impact has been the profound reduction in cold ischaemic time [16, 20]. Several studies have shown that a prolonged cold ischaemic time is more deleterious in recipients receiving kidney transplants from DCD donors than in recipients from DBD donors [21–23]. This finding, with DCD grafts being more 'vulnerable' to cold ischaemia than DBD grafts, has recently been confirmed by colleagues in The Netherlands [20], and might also explain why graft survival rates have improved to a greater extent for recipients of DCD donors than for recipients of DBD donors. Another possible explanation is that—as cold ischaemic periods of more than 24 hours in DCD kidneys are associated with worse graft survival—the proportion of procedures with excessive cold ischaemic periods (>24 hours) decreased over time with the increasing awareness of avoiding long cold ischaemic times in the Netherlands [4]. A potential contribution of hypothermic machine perfusion (HMP) to improved outcomes, however, is limited in this Dutch cohort since HMP was only fully implemented in the year 2016, and as the available data indicate that although HMP reduces the risk of DGF, it has a limited impact on the other outcome data [24].

Improved outcomes over time may further reflect advances in immunosuppressive therapies including the conversion from cyclosporine to tacrolimus as standard maintenance regime [25, 26]. Also, the introduction of more sensitive techniques to detect anti-human leukocyte antigen antibodies, such as the LUMINEX technique may have resulted in increased graft survival [18].

An alternative and non-exclusive explanation for the improved outcomes is the presence of a learning curve that involves the intangible and often intuitive aspects of medical decision-making processes in the context of donor and recipient selection, and organ allocation. Existence of a learning curve phenomenon is supported by the observation that outcomes of transplanted DBD kidneys improved significantly in a similar way as for DCD transplant procedures, but at an earlier (1990–1998) time era, and by the dynamics of the incidence of early graft loss over time (Fig 1). To be more specific, data for DBD procedures show a steep decline and stabilization of the incidence of early graft loss after 1998. A similar—albeit postponed—pattern is seen for the DCD procedures, for which the early graft loss incidence rate dropped and stabilized following 2008. Thus, it is likely that countries initiated a controlled DCD programme, may also experience some form of learning curve with transient inferior outcomes for this type of transplant procedure.

Remarkably, the time-related improvements in DCD outcomes occurred despite considerable increases in donor and recipient age, and in KDRI over time (Table 1). The apparent paradox of increasing KDRI values but improving graft survival rates suggests that, after the medical decision to accept a kidney for donation, there is a limited impact of donor characteristics on graft survival. This is illustrated by the remarkably low Nagelkerke $R^2$ for the association between KDRI and 5-year graft survival [13]. Hence, these data illustrate that graft survival reflects an interplay of donor, procedural and recipient factors [15].

This study has some limitations. Firstly, it is a registry-based study, which is associated with inherent design limitations. Secondly, this is a country-specific study as outcomes are influenced by national guidelines and decision-making policies. However, considering the liberal attitude towards DCD kidneys in The Netherlands (reflected by an equal distribution in DBD and DCD procedures, comparable donor ages and KDRI values) it is unlikely that the results reflect a high threshold in accepting DCD grafts.

In conclusion, this registry-based study shows a major improvement in outcomes of transplanted DCD kidneys over time, with DBD and DCD outcome equivalence in the current

timeframe. The time-related improvements for DCD outcomes not only show that DCD kidneys can be fully embraced, but also emphasise that careful interpretation is required for conclusions that are based on historical cohorts.

## Supporting information

**S1 Table. Multivariate analyses of posttransplant outcomes.** In total 16 multivariate analyses were performed. Each row represents a single multivariate analysis. The multivariate models were adjusted for variables that were statistically relevant in the univariate analysis (p-value <0.1). (-) Indicates that the variable was not included in the multivariate analysis. (*) p-value <0.05, (**) p-value <0.005. 95% CI, 95% confidence interval; DGF, delayed graft function; eGFR, estimated glomerular filtration rate; HR, hazard ratio; OR, odds ratio.
(DOCX)

## Acknowledgments

We thank the Dutch Transplant Foundation (Nederlandse Transplantatie Stichting) for providing the data.

## Author Contributions

**Conceptualization:** Michèle J. C. de Kok, Alexander F. M. Schaapherder, Aiko P. J. de Vries, Jan H. N. Lindeman.

**Formal analysis:** Michèle J. C. de Kok.

**Methodology:** Michèle J. C. de Kok, Alexander F. M. Schaapherder, Esther Bastiaannet, Jan H. N. Lindeman.

**Supervision:** Alexander F. M. Schaapherder, Ian P. J. Alwayn, Jan H. N. Lindeman.

**Writing – original draft:** Michèle J. C. de Kok, Alexander F. M. Schaapherder, Jan H. N. Lindeman.

**Writing – review & editing:** Alexander F. M. Schaapherder, Ian P. J. Alwayn, Frederike J. Bemelman, Jacqueline van de Wetering, Arjan D. van Zuilen, Maarten H. L. Christiaans, Marije C. Baas, Azam S. Nurmohamed, Stefan P. Berger, Esther Bastiaannet, Rutger J. Ploeg, Aiko P. J. de Vries, Jan H. N. Lindeman.

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
