## [Decision Letter · Decision Letter 0]

9 Jun 2020

PONE-D-20-14599

Improving outcomes for donation after circulatory death kidney transplantation: Science of the times.

PLOS ONE

Dear Dr. Lindeman,

Thank you for submitting your manuscript to PLOS ONE. After careful consideration, we feel that it has merit but does not fully meet PLOS ONE’s publication criteria as it currently stands. Therefore, we invite you to submit a revised version of the manuscript that addresses the points raised during the review process.

ACADEMIC EDITOR:

Interesting and important registry study on DCD kidney transplantation in the Netherlands, a high volume DCD KTx country. Two time periods have been compared, with a significant improvement in outcomes over time, which is indeed novel data.

Three expert reviewers have shown interest, but also recommended major revisions to the MS, and I agree with this assessment. They include concerns regarding methods, clarifications on data, the use of KDRI, the use of HMP, and discrepancies in the total numbers of DD transplants performed, along with additional queries.

These will need to be addressed meticulously, and I am looking forward to your revised MS.

We look forward to receiving your revised manuscript.

Kind regards,

Frank JMF Dor, M.D., Ph.D., FEBS, FRCS

Academic Editor

PLOS ONE

Journal Requirements:

2. In ethics statement in the manuscript and in the online submission form, please provide additional information about the database used in your retrospective study. Specifically, please ensure that you have discussed whether all data were fully anonymized before you accessed them and/or whether the IRB or ethics committee waived the requirement for informed consent. If patients provided informed written consent to have their data used in research, please include this information.

Reviewers' comments:

Reviewer's Responses to Questions

**Comments to the Author**

1. Is the manuscript technically sound, and do the data support the conclusions?

Reviewer #1: Partly

Reviewer #2: Yes

Reviewer #3: Yes

2. Has the statistical analysis been performed appropriately and rigorously? 

Reviewer #1: No

Reviewer #2: Yes

Reviewer #3: Yes

3. Have the authors made all data underlying the findings in their manuscript fully available?

Reviewer #1: No

Reviewer #2: Yes

Reviewer #3: Yes

4. Is the manuscript presented in an intelligible fashion and written in standard English?

Reviewer #1: Yes

Reviewer #2: Yes

Reviewer #3: Yes

5. Review Comments to the Author

Reviewer #1: In general, this is an important and interesting topic: it has been notable that transplantation from DCD donors has been common in the Netherlands, and the early post-transplant results have been poor historically. This is a large cohort study that demonstrates how outcomes have improved.

I found it very difficult to interpret the results for a few reasons:

1. How much data was missing? The authors describe imputing missing data points, but without some sense of the scale of the problem, it is impossible to comment on how appropriately this was done.

2. What variables were considered for inclusion in the multivariate models? What were actually included? What was the size of the effects of these variables?

3. I am unclear as to how KDRI was used. This includes both donor age (as the most important factor) and DCD/DBD status. How can you use KDRI to risk-adjust and then present data for DCD and DBD? Was there a double adjustment for donor age? (I may have mis-interpreted how it was used). Is KDRI appropriate for the local population?

4. The abstract contains no results. It would be helpful to have some numbers in the abstract.

Reviewer #2: The manuscript contains a register analysis. The register is the mandatory kidney transplant registry of the Netherlands. Therefore, the data completeness is granted. The question is whether kidney transplantation after donation from brain death donor (DBD) or cardiac defined death donor (DCD) have a difference in early graft loss and in particularly whether there were changes over time. Two periods of time were compared 1998-2007 and 2008-2017 with a total of 10307 deceased donor transplants, see below. The question is relevant and clear. The observation number is sufficient. The result is that there is no longer a difference between DBD and DCD concerning early graft loss during the second time period.

The clear-cut question gets an answer that is robust due to the large number of observations and rather simple and reliable statistics employed. The message is interesting information for people involved in kidney transplantation.

In the discussion the authors speculate on possible factors causing this improvement. The cold ischemia time is one of the mentioned mechanisms. In the data table the mean values for cold ischemia time are given for both periods. To further explore whether cold ischemia is a critical factor a multivariate analysis including the available parameters such as time period, cold ischemia time, kidney donor risk index KDRI, time on dialysis etc. might help to shed more light on this question.

A major critical point is the description of the observed transplantations. In the abstract 10307 deceased donor transplants between 1990 and 2018 were included. In material and methods section (line 90) it is 11415 deceased donor transplants between 1990 and 2018. However, the real observation periods were from 1/1998 to 12/2007 3499 transplants and from 1/2008 to 12/2017 3781 transplants, see table 1. This discrepancy between the total number of transplants has to be resolved.

In the abstract the numbers should be limited to the cases evaluated for the presented analysis.

Reviewer #3: the authors present a well-written analysis of post KT outcomes in the Netherlands. They compare DBD and DCD recipients in two sequential time cohorts and not a significant improvement in outcome especially for the DCD cohort. This is very interesting and has not previously been reported. I have the following comments.

1. The most significant difference is a nearly 5 hour reduction in cold ischemia time. I do think this is potentially a more important observation which I would recommend highlighting in the abstract, as well as somewhat more strongly in the discussion.

2. I would also like to know if there is any other technical change that occurred that could contribute to this-- like the type of preservation fluid in DCD, or use of heparin in DCD, or not using DCD donors who took more than a certain amount of time to expire (this is reflected in the first warm ischemia time which is slightly short in the recent time cohort-- was there an upper cut off instituted as a national standard?)

3. It is also not clear the use of kidney pumps-- I think this is what you mean by hypothermic machine perfusion (HMP) but I also know there is a practice for some European centers to pump the donor via ECMO, so could you clarify if kidneys are routinely place on pumps after organ recovery, and if so, did this really only start in 2016 and is it applied only for high KDRI or only for DCD.

6. PLOS authors have the option to publish the peer review history of their article (what does this mean?). If published, this will include your full peer review and any attached files.

Reviewer #1: No

Reviewer #2: No

Reviewer #3: No

---

## [Author Response · Author response to Decision Letter 0]

18 Jun 2020

Academic editor

Interesting and important registry study on DCD kidney transplantation in the Netherlands, a high volume DCD KTx country. Two time periods have been compared, with a significant improvement in outcomes over time, which is indeed novel data. 

Three expert reviewers have shown interest, but also recommended major revisions to the MS, and I agree with this assessment. They include concerns regarding methods, clarifications on data, the use of KDRI, the use of HMP, and discrepancies in the total numbers of DD transplants performed, along with additional queries.

These will need to be addressed meticulously, and I am looking forward to your revised MS.

Thank you for reviewing our work and your valuable comments, which have helped us to improve the manuscript. A detailed response to your and the reviewers’ comments, as well as the revised manuscript, can be found below. 

Reviewer #1

In general, this is an important and interesting topic: it has been notable that transplantation from DCD donors has been common in the Netherlands, and the early post-transplant results have been poor historically. This is a large cohort study that demonstrates how outcomes have improved. 

We thank the reviewer for his or her supportive comments.

I found it very difficult to interpret the results for a few reasons:

1. How much data was missing? The authors describe imputing missing data points, but without some sense of the scale of the problem, it is impossible to comment on how appropriately this was done

We agree with the reviewer that the text is unclear with respect to this point. Multiple imputation was only performed for the KDRI in order to facilitate comparison of the Dutch donor cohort with that of other countries for the 1998-2018 interval. However, donor hypertension and diabetes are only registered in The Dutch National Organ Transplant Registry from 2000 respectively 2002 onwards. As such, the proportion of missing data (26.6% for diabetes, and 10.0% for hypertension) prompted our decision to apply multiple imputation for the variables included in the KDRI. To better illustrate the scale of missing data, we have now expanded the information in the methods section (Methods page 6, line 124-129).

2. What variables were considered for inclusion in the multivariate models? What were actually included? What was the size of the effects of these variables?

These are important questions and we agree with the reviewer that this information is missing. Variables included in the multivariate analyses, were all found statically relevant in the univariate analysis (p-value <0.1) (Methods page 7, line 133-134). In total, 16 multivariate models were performed, and following the reviewer’s advice we have now included a table illustrating the variables included in the multivariate models and their respective effect sizes (S1 Table).

3. I am unclear as to how KDRI was used. This includes both donor age (as the most important factor) and DCD/DBD status. How can you use KDRI to risk-adjust and then present data for DCD and DBD? Was there a double adjustment for donor age? (I may have mis-interpreted how it was used). Is KDRI appropriate for the local population?

We apologize for being unclear. In this study, we presented the imputed-KDRI in Table 1 (Baseline characteristics) to facilitate comparison of the Dutch donor cohort with that of other countries. We have deliberately chosen to exclude the KDRI in the multivariate models in order to avoid overcorrection (as the KDRI also comprises the variables donor age and donor type which are already included separately in the univariate and multivariate analyses). This has now been clarified in the methods section (Methods page 7, line 134-136).

With regard to the last question, whether the KDRI is appropriate for the local population, is an interesting question and has been previously investigated. In fact, the KDRI as proposed by Rao et al. was externally validated in the Dutch cohort and the authors concluded that the KDRI performs equally well in the Dutch population [Peters-Sengers et al]. As such, we consider the KDRI appropriate for the local population. 

Peters-Sengers H, et al. “Validation of the Prognostic Kidney Donor Risk Index Scoring System of Deceased Donors for Renal Transplantation in the Netherlands.” Transplantation vol. 102,1 (2018): 162-170.

4. The abstract contains no results. It would be helpful to have some numbers in the abstract.

We agree with the reviewer that it would be helpful to have some numbers in the abstract. With the permitted number of words in mind, we have tried to maximize the information in the abstract. (Abstract page 2, line 34, 42-45).

Reviewer #2

The manuscript contains a register analysis. The register is the mandatory kidney transplant registry of the Netherlands. Therefore, the data completeness is granted. The question is whether kidney transplantation after donation from brain death donor (DBD) or cardiac defined death donor (DCD) have a difference in early graft loss and in particularly whether there were changes over time. Two periods of time were compared 1998-2007 and 2008-2017 with a total of 10307 deceased donor transplants, see below. The question is relevant and clear. The observation number is sufficient. The result is that there is no longer a difference between DBD and DCD concerning early graft loss during the second time period. The clear-cut question gets an answer that is robust due to the large number of observations and rather simple and reliable statistics employed. The message is interesting information for people involved in kidney transplantation.

The reviewer’s supportive comments are highly appreciated. 

1. In the discussion the authors speculate on possible factors causing this improvement. The cold ischemia time is one of the mentioned mechanisms. In the data table the mean values for cold ischemia time are given for both periods. To further explore whether cold ischemia is a critical factor a multivariate analysis including the available parameters such as time period, cold ischemia time, kidney donor risk index KDRI, time on dialysis etc. might help to shed more light on this question.

We agree with the reviewer that this needs further elaboration. To address this point, we have now included a table that includes all multivariate models which also includes the effect of cold ischaemia time on outcome (S1 Table). The KDRI is not presented in this table as we have deliberately chosen to exclude this variable from the multivariate analyses to avoid overcorrection. This relevant point has also been addressed in question 2 of reviewer 1, and has been clarified in the methods section (Methods page 7, line 134-136).

2. A major critical point is the description of the observed transplantations. In the abstract 10307 deceased donor transplants between 1990 and 2018 were included. In material and methods section (line 90) it is 11415 deceased donor transplants between 1990 and 2018. However, the real observation periods were from 1/1998 to 12/2007 3499 transplants and from 1/2008 to 12/2017 3781 transplants, see table 1. This discrepancy between the total number of transplants has to be resolved.

We truly apologize for being unclear. We have now clarified the numbers of included patients in the methods section (Methods page 5, line 90 and 95-101). 

3. In the abstract the numbers should be limited to the cases evaluated for the presented analysis.

We agree with the reviewer that the number of cases evaluated in this study should be presented in the abstract. We have therefore added this information in the abstract (Abstract page 2, line 33-35).

Reviewer #3

The authors present a well-written analysis of post KT outcomes in the Netherlands. They compare DBD and DCD recipients in two sequential time cohorts and not a significant improvement in outcome especially for the DCD cohort. This is very interesting and has not previously been reported. 

We thank the reviewer for these valuable comments.

I have the following comments.

1. The most significant difference is a nearly 5 hour reduction in cold ischemia time. I do think this is potentially a more important observation which I would recommend highlighting in the abstract, as well as somewhat more strongly in the discussion.

We fully agree with the author that this is an important observation and can be stated more clearly in the abstract and the discussion. In order to emphasize this, we have now added this information in the abstract (Abstract page 2, line 44-45) and expanded the discussion section (Discussion page 14, line 233-236). To be more specific, another possible explanation for improved outcomes over time is that - as cold ischaemic periods of more than 24 hours in DCD kidneys are associated with worse graft survival [Schaapherder et al.]- the proportion of procedures with excessive cold ischaemia times (>24 hours) decreased over time with the increasing awareness of avoiding long cold ischaemic times in the Netherlands. In order to emphasize this, we have added this information in the baseline characteristics table, results section and discussion section (Table 1; Results page 8, line 170-172; Discussion page 14, line 233-236). 

Schaapherder, A et al. “Equivalent Long-term Transplantation Outcomes for Kidneys Donated After Brain Death and Cardiac Death: Conclusions From a Nationwide Evaluation.” EClinicalMedicine vol. 4-5 25-31. 9 Oct. 2018). 

2. I would also like to know if there is any other technical change that occurred that could contribute to this-- like the type of preservation fluid in DCD, or use of heparin in DCD, or not using DCD donors who took more than a certain amount of time to expire (this is reflected in the first warm ischemia time which is slightly short in the recent time cohort-- was there an upper cut off instituted as a national standard?). 

The reviewer brings up an interesting point. In the Eurotransplant organ donation protocol, the DBD donor receives heparin intravenously before the start of the cold perfusion. In the case of the DCD donor, heparin is added to the preservation fluid. This aspect of the protocol has not changed over time.

The reviewer is right that there is a national upper cut-off in the context of kidney donation. To be more specific, in The Netherlands, the time from withdrawal of life sustaining therapy to circulatory death (also known as the agonal period) has been limited to 2 hours for kidney donation, and this maximum time period did not change over time. Although it would be interesting to compare the mean values over time, this is not possible as the agonal time is not registered in The Dutch National Organ Transplant Registry. Furthermore it should be noted that the agonal period is not reflected in the first warm ischaemic period, as the first warm ischaemic period begins after the agonal phase and no touch period (Methods page 6, line 112-114). 

Fortunately, the type of preservation fluid is included in The Dutch National Organ Transplant Registry and indeed there is a change in type of preservation fluid in DCD donor kidneys over time. This aspect has now been included in Table 1, the results and discussion (Table 1; Results page 9, line 173-176; and Discussion page (results page 8-9, line 162-165) and discussion (discussion, page 13, line 225). We have deliberately chosen not to include the type of preservation in the univariate and multivariate analyses, as this variable is collinear (the selection of preservation solution depends on donor type) and would therefore impact the validity of the model (Methods page 7, line 136-139) 

3. It is also not clear the use of kidney pumps-- I think this is what you mean by hypothermic machine perfusion (HMP) but I also know there is a practice for some European centers to pump the donor via ECMO, so could you clarify if kidneys are routinely place on pumps after organ recovery, and if so, did this really only start in 2016 and is it applied only for high KDRI or only for DCD.

It is correct that hypothermic machine perfusion (HMP) preserves kidney grafts after retrieval (i.e. ex-vivo) by administration of cold preservation solution with the use of a pump. In The Netherlands, the use of HMP is implemented from the year 2016 onwards for all donated donor kidneys regardless of the donor type or donor quality (KDRI). In previous years, HMP was only incidentally performed in the research setting. 

Preservation of organs with normothermic regional perfusion (NRP), which is based on the use of ECMO devices, is performed in the donor (in-vivo) and applied for the first time in October 2018 in the Netherlands in research setting. As such the number of transplantation performed after NRP is neglectable in this study population (1990-2018). Nevertheless, we agree that it would definitely be interesting to evaluate the effect of NRP on outcomes in the future.

---

## [Decision Letter · Decision Letter 1]

13 Jul 2020

Improving outcomes for donation after circulatory death kidney transplantation: Science of the times.

PONE-D-20-14599R1

Dear Dr. Lindeman,

We’re pleased to inform you that your manuscript has been judged scientifically suitable for publication and will be formally accepted for publication once it meets all outstanding technical requirements.

Kind regards,

Frank JMF Dor, M.D., Ph.D., FEBS, FRCS

Academic Editor

PLOS ONE

Additional Editor Comments (optional):

Reviewers' comments:

Reviewer's Responses to Questions

**Comments to the Author**

1. If the authors have adequately addressed your comments raised in a previous round of review and you feel that this manuscript is now acceptable for publication, you may indicate that here to bypass the “Comments to the Author” section, enter your conflict of interest statement in the “Confidential to Editor” section, and submit your "Accept" recommendation.

Reviewer #1: All comments have been addressed

Reviewer #2: All comments have been addressed

2. Is the manuscript technically sound, and do the data support the conclusions?

Reviewer #1: Yes

Reviewer #2: Yes

3. Has the statistical analysis been performed appropriately and rigorously? 

Reviewer #1: Yes

Reviewer #2: Yes

4. Have the authors made all data underlying the findings in their manuscript fully available?

Reviewer #1: No

Reviewer #2: Yes

5. Is the manuscript presented in an intelligible fashion and written in standard English?

Reviewer #1: Yes

Reviewer #2: Yes

6. Review Comments to the Author

Reviewer #1: Thank you for addressing my comments. The manuscript is much clearer and more explicit about methodology.

Reviewer #2: (No Response)

7. PLOS authors have the option to publish the peer review history of their article (what does this mean?). If published, this will include your full peer review and any attached files.

Reviewer #1: No

Reviewer #2: **Yes: **Dirk L. Stippel

---

## [Editor Report · Acceptance letter]

15 Jul 2020

PONE-D-20-14599R1 

Improving outcomes for donation after circulatory death kidney transplantation: Science of the times. 

Dear Dr. Lindeman:

I'm pleased to inform you that your manuscript has been deemed suitable for publication in PLOS ONE. Congratulations! Your manuscript is now with our production department. 

Kind regards, 

on behalf of

Dr. Frank JMF Dor 

Academic Editor

PLOS ONE